# New Fully Automated Preparation of High Apparent Molar Activity ^68^Ga-FAPI-46 on a Trasis AiO Platform

**DOI:** 10.3390/molecules27030675

**Published:** 2022-01-20

**Authors:** Chiara Da Pieve, Marta Costa Braga, David R. Turton, Frank A. Valla, Pinar Cakmak, Karl-Heinz Plate, Gabriela Kramer-Marek

**Affiliations:** 1Preclinical Molecular Imaging, Division of Radiotherapy and Imaging, The Institute of Cancer Research, London SW7 3RP, UK; marta.costabraga@icr.ac.uk (M.C.B.); DavidRobert.Turton@icr.ac.uk (D.R.T.); 2Sofie, Dulles, VA 20166, USA; Frank.Valla@sofie.com; 3Institute of Neurology (Edinger Institute), University Hospital Frankfurt, 60590 Frankfurt, Germany; pinar.cakmak@kgu.de (P.C.); Karlheinz.Plate@kgu.de (K.-H.P.); 4German Consortium for Translational Cancer Research (DKTK), Partner Site Frankfurt/Mainz, 60590 Frankfurt, Germany; 5Frankfurt Cancer Institute, 60590 Frankfurt, Germany

**Keywords:** FAP, gallium-68, FAPI-46, automated radiosynthesis, Trasis AiO, preclinical PET

## Abstract

A large number of applications for fibroblast activation protein inhibitors (FAPI)-based PET agents have been evaluated in conditions ranging from cancer to non-malignant diseases such as myocardial infarction. In particular, ^68^Ga-FAPI-46 was reported to have a high specificity and affinity for FAP-expressing cells, a fast and high accumulation in tumor lesions/injuries together with a fast body clearance when investigated in vivo. Due to the increasing interest in the use of the agent both preclinically and clinically, we developed an automated synthesis for the production of ^68^Ga-FAPI-46 on a Trasis AiO platform. The new synthetic procedure, which included the processing of the generator eluate using a strong cation exchange resin and a final purification step through an HLB followed by a QMA cartridge, yielded ^68^Ga-FAPI-46 with high radiochemical purity (>98%) and apparent molar activity (271.1 ± 105.6 MBq/nmol). Additionally, the in vitro and in vivo properties of the product were assessed on glioblastoma cells and mouse model. Although developed for the preparation of ^68^Ga-FAPI-46 for preclinical use, our method can be adapted for clinical production as a reliable alternative to the manual (i.e., cold kit) or modular systems preparations already described in the literature.

## 1. Introduction

The membrane-associated protease fibroblast activation protein (FAP) is upregulated in the stroma of a large variety of cancers [1,2] and in inflammatory conditions such as liver cirrhosis and both cardiovascular and rheumatoid diseases [3,4]. In cancer, FAP overexpression is associated with high tumor stage/grade, lymph node invasion, recurrence and reduced patient survival [5]. Consequently, FAP is an attractive target for diagnostic and therapeutic purposes [6].

Amongst the molecules developed to inhibit FAP for therapeutic purposes, a group of quinoline-based compounds showed great potential [7]. In particular, a DOTA-bearing quinoline-based structure (FAPI-46) was labeled with the PET radionuclide gallium-68 for imaging and with lutetium-177 for radionuclide therapy [6,8]. Due to ^68^Ga-FAPI-46′s high lesion uptake, rapid body clearance and low accumulation in normal tissues, this agent has been investigated in a large number of patients with different tumor entities and also non-malignant diseases [9].

The straightforward preparation of gallium-68-radiolabeled FAPIs, for both preclinical and clinical use, is mainly carried out manually, with the prospect to develop standardizable cold-kits for their cGMP production for clinical applications [10,11]. However, the increasing demand of ^68^Ga-FAPI-46, as for radiotracers and radiopharmaceuticals in general, cannot be fulfilled by manual production alone. The development of automated radiosynthesis procedures is hence necessary to meet production, regulatory and radioprotection requirements. So far, few automated modular systems have been used and investigated for the synthesis of gallium-68-labeled FAPIs. Varasteh et al. prepared ^68^Ga-FAPI-04 using a Scintomics GallElut + module to investigate the post-myocardial infarction (MI) fibroblast activation in a MI rat model [12]. Additionally, Spreckelmeyer et al. evaluated the fully automated synthesis of ^68^Ga-FAPI-46 for clinical applications on the commercially available Modular Lab PharmTracer and ML eazy modules [13].

Our study provides a detailed fully automated procedure for the production of ^68^Ga-FAPI-46 on the extensively used commercial cassette-based synthesis platform Trasis AllinOne (AiO). The development of robust and reproducible synthetic methods on a wide variety of automated systems is a valuable tool to extend the availability and production of FAPI based radiopharmaceuticals for both preclinical and clinical applications. Additionally, the new synthetic method yielded a product with reproducibly good radiochemical purity (>98%), stability and high apparent molar activity (213 ± 71 MBq/nmol) owing to the use of a combination of a polymeric-based and a strong anion exchange SPE cartridge for the purification of the final product. The radioligand performance was further tested in vitro showing high sensitivity and specificity to the target. Furthermore, increased accumulation of the agent in FAP-positive subcutaneous glioblastoma (U87-MGvIII) tumors allowed the acquisition of high contrast PET images.

The novel automatic method was mainly developed as an alternative to the manual synthesis to fulfil the high demand of the agent for preclinical evaluation in tumor mice models and to avoid excess exposure of the operator to radiation. However, after optimizing the purity profile of the final product and determining the whole range of quality control parameters, the described process could be adapted for the cGMP preparation of ^68^Ga-FAPI-46 for patient use.

## 2. Results and Discussion

### 2.1. Automated Radiolabeling Using the Trasis AiO Platform

Due to the high positron energy of ^68^Ga, the manual radiolabeling process (in particular, crucial steps like the post-processing of the generator eluate and the final product purification) results in high hand doses to the operator. A fully automated method would reduce the exposure of the user to radiation due to the absence of direct intervention. Furthermore, an automated procedure will have greater control over reaction times, temperatures and flow rates and could be quicker and more reproducible than the manual approach. Hence, the preparation of ^68^Ga-FAPI-46 (Appendix A) was automated on a Trasis AiO platform using a cassette assembled as shown in Figure 1. In brief, the ^68^Ga generator eluate was purified and concentrated using a strong cation exchange cartridge (SCX) and buffered to pH 4 using 0.2 M sodium acetate. FAPI-46 was then added to the ^68^Ga solution (350–900 MBq, ca. 1 mL). The mixture was incubated at 95 °C for 10 min before being purified by solid phase extraction (SPE). The production time was approximately 35 min. The decay corrected radiochemical yield was 66.0 ± 7.6% and the apparent molar activity (AMA) was 271.1 ± 105.6 MBq/nmol. The SCX column used at the beginning of the process both reduces the volume of the ^68^Ga eluate (from ca. 1.1 mL to 0.4 mL) and removes the metal impurities (including ^68^Ge^4+^ and Zn^2+^) from the solution. A higher activity concentration and a reduction in the levels of interfering metal impurities would improve the radiolabeling yield and the product’s molar activity [14]. A summary of the automated procedure is shown in Table 1.

### 2.2. Purification by SPE

The purification of the final product was performed by simple SPE using one HLB column followed by one QMA cartridge (HLB-QMA combination). The HLB (hydrophilic lipophilic balanced) cartridge contains a polymeric solid phase extraction material that is able to retain a wide range of compounds with different polarities. Additionally, its use allowed the recovery of the product in a small volume (approximately 400–500 µL). This resulted in a maximized activity concentration (i.e., the activity per volume, 390.6 ± 215.6 MBq/mL at the end of synthesis), an important factor when only a small volume of agent can be injected into small rodents for imaging purposes. The QMA cartridge contains a silica based strong anion exchange sorbent and can be used in both aqueous and non-aqueous solutions. As QMA is an anion exchanger, it will retain negatively charged compounds, such as the FAPI-46 precursor which has free carboxylic acid groups. The HLB-QMA combination yielded ^68^Ga-FAPI-46 with a RCP > 98% and efficiently removed most of the UV active impurities present in the reaction mixture (Figure 2 and Appendix A). Conversely, although having a very high RCP, the products achieved by using either the HLB cartridge alone (Appendix A) or a QMA-HLB combination (i.e., a QMA cartridge followed by an HLB column) contained a considerable amount of un-known UV active species (Appendix A). This is suggesting that the order in which the reaction mixture passes through the cartridges pair is crucial to achieve an improved purification. When the reaction solution is loaded onto the HLB cartridge, ^68^Ga-FAPI-46 is retained while polar molecules (including ascorbic acid, acetate from the buffer, as well as free ^68^Ga^3+^) are washed off to waste. The use of an ethanol-water solution displaces ^68^Ga-FAPI-46 from the HLB cartridge with impurities of similar polarity. Once this solution is loaded onto the QMA cartridge, acidic and negatively charged molecules in general (such as the FAPI-46 precursor) are retained by the stationary phase while neutral components such as ^68^Ga-FAPI-46 pass thought the cartridge unretained and are collected into the product vial. It is possible that for the QMA cartridge to work effectively, the number of anionic compounds (such as ascorbic acid and buffers constituents) in the solution should be as small as possible and they should be removed from the product stream prior to the cartridge use. The presence of large quantities of negatively charged molecules in solution is probably the reason behind the low purification efficiency when the QMA is used before the HLB cartridge. A molar activity (i.e., the activity per ligand, MBq/nmol) in the region of 2700–6390 MBq/nmol was calculated (using a calibration curve and considering just the area of the UV peak of the product). However, a variety of UV absorbing material was present in the final product solution (Figure 2B and Appendix A). Assumed to derive from the precursor, these impurities could bind to FAP and were included in the molar activity calculation. The AMA of the purified product was calculated taking into consideration the contribution of all UV active substances in the HPLC chromatogram (Appendix A), presuming they have similar molar extinction coefficients. Based on this assumption, the product derived from the HLB-QMA cartridges combination had a considerably higher AMA (271.1 ± 105.6 MBq/nmol at the end of synthesis) compared to the other two tested SPE alternatives (13.6 ± 1.5 MBq/nmol and 17.5 MBq/nmol for HLB only and QMA-HLB, respectively). Additionally, for our experiments, we used the eluate from two generators of different age (two and eight months old). Reduced ^68^Ga activities and growing metallic impurities quantities in the eluate are generally associated with the increased age of generators and can affect the radiolabeling efficiency (expressed as molar activity of the final product). The recorded apparent molar activity values were directly linked to the age of the generator, and therefore to the quality of the eluate, with the higher (366–432.9 MBq/nmol) and lower (131–297 MBq/nmol) apparent molar activities being achieved using ^68^Ga from a new and an older generator, respectively.

The molar (as well as the specific) activity of a radioactive agent is a very important variable for in vitro and preclinical in vivo applications. In general, high molar activity is crucial when only a small amount of a radioactive compound can be injected for the imaging of the target (e.g., receptor, enzyme, cell membrane associated protein), especially if it is present in low abundance. More importantly, the use of a standardized radioactive and non-radioactive quantity of the agent is advisable for consistent, reproducible and comparable results. For that reason, an optimal molar activity may have to be determined experimentally for each particular application by starting from a high molar activity product and adding appropriate quantities of the non-radioactive compound. The high molar activity of the ^68^Ga-FAPI-46 produced using our procedure enables this evaluation. Moreover, the same batch could be used over a longer period of time guaranteeing a product with a sufficient and consistent molar activity at the time of use.

As usually recommended, a radioprotectant such as ascorbic acid was used during the radiolabeling reaction [15]. It was added also to the purified product solution, as some degree of radiolysis was observed (Appendix A).

Due to the purification method, from the present synthetic process we expect a product with superior chemical purity compared to other reported methods. Although it was developed to produce ^68^Ga-FAPI-46 suitable for the preclinical use (i.e., with high AMA and activity concentration), our method can be adapted for the clinical production. Importantly, due to the availability of GMP produced pre-packaged disposable cassette and reagents, a GMP compliant production of ^68^Ga-FAPI-46 on a Trasis AiO platform can be achieved. In that case, the optimization of factors such as the quantity of precursor, the elution of the product from the cartridges (e.g., volume and composition of the eluent), the final product formulation and consequent stability for the determination of its shelf-life should be carried out as well as the chemical profile characterization of the final product.

### 2.3. Binding and Internalization Studies

To test the quality of the ^68^Ga-FAPI-46 produced using our automated method and to confirm the ability to target FAP with high sensitivity and specificity, we incubated the agent with cell lines having high (SF-539), medium (U87-MGvIII) and low (A549) FAP expression. ^68^Ga-FAPI-46 recognized the target with high sensitivity and specificity. As shown in Figure 3A, the highest binding was seen in the SF-539 cells (6.2 ± 0.4 %ID/mg), followed by U87-MGvIII (2.1 ± 0.3 %ID/mg). No significant binding was observed in FAP-negative A549 cells (0.2 ± 0.03 %ID/mg). The measured cell-associated radioactivity was in agreement with the cellular FAP expression determined by flow cytometry (Figure 3B). Furthermore, pre-incubating the cells with 100-fold molar excess of non-radioactive ^nat^Ga-FAPI-46 significantly reduced the radioactivity signal in the positive cell lines (*p* < 0.001 for SF-539 and *p* < 0.01 for U87-MGvIII) further confirming the binding specificity of ^68^Ga-FAPI-46 to the target. Additionally, the internalization of ^68^Ga-FAPI-46, assessed using SF-539 cells, was fast. Of note, over 80% of all cell-bound radioactivity was found inside the cell already after 15 min (Figure 3C). These results are consistent with previously published data reporting similar radiolabeled FAPI agents [8].

### 2.4. In Vivo Studies

The targeting properties of ^68^Ga-FAPI-46 were investigated using U87-MGvIII-tumor bearing mice. High contrast PET/CT images were acquired as early as 30 min post-^68^Ga-FAPI-46 injection (2 MBq, 65 pmol per mouse, apparent molar activity of 30.8 MBq/nmol) (Figure 4). The tumor uptake, derived from PET images, had a SUV_mean_ of 0.58 ± 0.09 at 30-45 min p.i. (Figure 4A). Immediately after PET/CT scans, the mice were euthanized and the major organs dissected and measured using a γ-counter. The biodistribution data corroborated the imaging studies and the high tumor uptake of ^68^Ga-FAPI-46 (6.41 ± 1.11 %ID/g) confirmed the ability of the agent to target tumors with medium FAP expression. In addition, the radioactivity in non-targeted organs such as the blood, heart, liver, spleen, large intestine and muscle was very low (Appendix A). The high tumor-to-organ ratios (tumor-to-blood: 2.8 ± 0.9; tumor-to-muscle: 3.6 ± 1.9; tumor-to-liver: 11.4 ± 3.2) enabled capturing high contrast PET images. A high bone uptake was detected (3.83 ± 1.19 %ID/g) (Appendix A) similarly to previously reported studies using ^68^Ga labeled FAPI derivatives [16]. This is possibly due to presence of FAP in both murine and human multipotent bone-marrow stem/stromal cells and mesenchymal stromal cells that reside in the bone [17,18,19]. The clearly identifiable bladder on the PET images (Figure 4A) together with the high kidney uptake (2.45 ± 0.94 %ID/g, Appendix A) indicate renal clearance and urinary excretion of the agent. The tumor targeting by ^68^Ga-FAPI-46 was associated with FAP expression assessed by immunohistochemistry staining (Figure 4B). Furthermore, the intense CD31 staining confirmed the presence of vascular endothelial cells and Ki67 staining reflected the tumor cell proliferation.

## 3. Material and Methods

### 3.1. General Materials and Methods

Chemicals and solvents were purchased and used without further purification unless otherwise stated. Ethanol (EtOH) and HPLC grade acetonitrile were purchased from Fisher Scientific (Loughborough, UK). Sodium acetate (AnalR Normapur) was purchased from VWR International (Lutterworth, UK). Sodium ascorbate, ascorbic acid and sodium chloride (NaCl) were obtained from Sigma-Aldrich (Gillingham, UK). The sodium chloride/hydrochloric acid (NaCl/HCl) solution was prepared from 5 M NaCl and 5.5 M HCl stock solutions as reported in the literature [20]. The FAPI-46 and the reference standard ^nat^Ga-FAPI-46 were kindly supplied by SOFIE Biosciences (Dulles, VA, USA). Oasis HLB (1 mL, 30 mg sorbent) and Sep-Pak Accell Plus QMA Plus Light (130 mg sorbent) SPE cartridges were purchased from Waters (Elstree, UK). Macherey-Nagel Chromabond SA cartridges (strong cation exchange SCX; 1 mL, 100 mg sorbent) were purchased from Fisher Scientific (Loughborough, UK). ^68^Gallium was eluted in 0.1 M hydrochloric acid from a Galli Ad ^68^Ge/^68^Ga generator (IRE ELiTE Radiopharma, Fleurus, Belgium) and supplied by the Royal Marsden Hospital radiopharmacy (Sutton, UK). The automated radiosynthesis platform used in the study was a Trasis AllinOne (AiO)™ (Trasis, Ans, Belgium) equipped with a 4 mL COC V-shaped reactor and a copper heater insert (Trasis, Ans, Belgium). Plastic syringes (2, 5 and 10 mL) and 100 Sterican 0.98 × 70 mm (20G × 2¾″) needles were purchased from Terumo Pharmaceutical Solutions (Medisave, Weymouth, UK) and BBraun (Hessen, Germany), respectively. Analytical RP-HPLC was carried out on an Agilent Infinity 1260 quaternary pump system equipped with a 1260 Diode array (Agilent Technologies, Didcot, UK) and a Scan Ram NaI radioHPLC detector (Lablogic, Sheffield, UK). Chromatographic separations were performed on a Luna C18 column, 4.6 × 150 mm, 5 µm (Phenomenex, Macclesfield, UK) by isocratic elution using 85% 10 mM NaH_2_PO_4_ and 15% acetonitrile as mobile phase at 1 mL/min flow rate. The UV absorbance was measured at the wavelength of 260 nm. Elution profiles were analyzed using Laura software v.4.4.6.79 (Lablogic, Sheffield, UK). Retention times (R_t_) are expressed as minutes:seconds (min:s). Instant thin layer chromatography analysis (ITLC) was performed using ITLC-SG strips (Agilent Technologies, Didcot, UK) and 75% methanol 25% 5 M ammonium acetate pH 7 as mobile phase. The ITLC strips were analyzed on a Cyclone Plus Phosphor Imager (Perkin Elmer, Beaconsfield, UK).

### 3.2. Trasis AiO Platform Setup and Preparation of ^68^Ga-FAPI-46

Before starting the radiolabeling process, the SPE cartridges were manually pre-conditioned with ethanol followed by 0.1 M HCl (SCX cartridge) or ethanol followed by water (HLB cartridge). The cassette (with pre-conditioned SPE cartridges) was connected to the Trasis AiO platform and a leak test was performed. The cassette was then loaded with the reagents. The cassette was structured as follows: the SCX cartridge for the concentration and purification of the ^68^Ga eluate was placed in position 5. The reservoir vial with the NaCl/HCl mixture used for the elution of ^68^Ga from the SCX cartridge was placed in position 3. The substrate solution (15 nmol, 15 µL of 1 mM solution of FAPI-46 in water) in 0.2 M sodium acetate pH 6 (0.6 mL) and sodium ascorbate (1 mg) was placed in position 8. A water reservoir used for loading and washing the HLB cartridge was attached to position 12. The HLB and the QMA cartridges were placed in position 17 and 18, respectively. The reservoir vial containing an ethanol-water solution (1:1 *v*/*v*) was placed in position 14. As the ^68^GaCl_3_ solution (i.e., ^68^Ge/^68^Ga generator eluate) was delivered in a vial from a remote facility, a ^68^GaCl_3_ solution-drawing step was used in the sequence. Before starting the automated process, the ^68^GaCl_3_ solution (approximately 1.1 mL) was manually diluted with 0.1 M HCl (1 mL) to maximize the transfer of the radioactive stock from the delivery vial to the cassette. The ^68^GaCl_3_ vial was connected to the Trasis AiO input line equipped with a needle reaching the bottom of the vial (position 1). The automated process was then started by drawing up the ^68^GaCl_3_ solution (approximately 2 mL) from the delivery vial into the syringe in position 2. The ^68^GaCl_3_ solution was pushed through the SCX cartridge (position 5). The cartridge was flushed with a flow of nitrogen. The NaCl/HCl mixture (0.5 mL) was drawn up from the reservoir in position 3 into the syringe in position 2. The NaCl/HCl mix was pushed through the SCX cartridge to elute the ^68^Ga into the reactor followed by a flow of nitrogen to complete the transfer. The precursor and ascorbic acid solution in 0.2 M sodium acetate pH 6 was added into the reactor from the syringe in position 8. The reactor was heated to 95 °C for 10 min. The reaction solution was then cooled with compressed air to 40 °C. From the vial in position 10, 1 mL of a 1 M sodium acetate buffer pH 4.5 was drawn up in the syringe in position 9 and was added to the reactor. The whole solution was then taken up into the syringe in position 9. The solution was diluted with water (6 mL), taken from the water reservoir in position 12, and slowly loaded onto the HLB cartridge at position 17. The reactor was rinsed with water (5 mL) and the solution was loaded onto the HLB cartridge. The liquid from the HLB cartridge output was sent to waste (position 6). The syringe (position 9) was rinsed again with water (from reservoir in position 12) to waste. The HLB cartridge was washed with an extra 2 mL of water from the syringe in position 9 and flushed with a flow of nitrogen to waste. The product was eluted from the HLB cartridge with an ethanol-water solution (1:1 *v*/*v*, 0.5 mL taken from the vial in position 14) using the syringe in position 15. The eluate from the HLB cartridge was directly passed through the QMA cartridge (position 18) and into the product vial that contained ascorbic acid (1 mg). The HLB and QMA cartridge were flushed with a flow of nitrogen to complete the product transfer. The purified product was analyzed by RP-HPLC and ITLC (Appendix A) and the radiochemical purity (RCP), the radiochemical yield (RCY) and the apparent molar activity (AMA) were determined. Synthesis time (from the beginning of the reaction) = approximately 35 min; analytical RP-HPLC: R_t_: 8:11 min:s; RCP > 98%; RCY (decay corrected at the beginning of reaction): 66.0 ± 7.6 % (n = 10); AMA (at end of reaction): 271.1 ± 105.6 MBq/nmol (n = 9).

### 3.3. Cell Culture and Characterization of FAP Expression

The human GBM cell line U87-MGvIII was kindly provided by Dr Frank Furnari (Ludwig Institute for Cancer Research, San Diego, CA, USA) and maintained as previously described [21]. The human non-small cell lung cancer cell line A549 and the human gliosarcoma cell line SF-539 were a courtesy of Udai Banerji and Igor Vivanco (The Institute of Cancer Research, London, UK), respectively. The A549 cell line was cultured in DMEM (Gibco, Thermo Fisher Scientific, Loughborough, UK) and the SF-539 in RPMI (Gibco, Thermo Fisher Scientific, Loughborough, UK), supplemented with 10% heat-inactivated fetal bovine serum (FBS, Gibco, Thermo Fisher Scientific, Loughborough, UK). All cells were maintained at 37 °C in a humidified atmosphere containing 5% CO_2_.

FAP expression was evaluated by flow cytometry. All cell lines (5 × 10^5^ cells/sample) were washed once with ice-cold phosphate buffered-saline (PBS, Gibco, Thermo Fisher Scientific, Loughborough, UK) and incubated for 1 h at RT with 5 µL/sample of either human FAP PE-conjugated antibody (FAB3715P-025, Bio-techne, Abingdon, UK) or PBS for unstained controls. After incubation, the cells were washed twice with ice-cold PBS and resuspended in 500 µL of PBS for data acquisition. Flow cytometry was performed using a BD^TM^ LSRII flow cytometer (BD Biosciences, Swindon, UK). A total of 10,000 events per sample was recorded and the population corresponding to single, alive cells was gated and shown as a histogram plot using FlowJo (BD, Biosciences, Swindon, UK). FAP expression was determined as the difference of mean fluorescence intensity (MFI) between stained and unstained samples.

### 3.4. In Vitro Studies

For the binding assay, cells (5 × 10^5^) were plated on 12-well plates 24 h prior to the experiment. On the day of the experiment, media were removed, the cells were washed with ice-cold PBS, and incubated with ^68^Ga-FAPI-46 (2 pM, ca. 250 kBq/well) for 1 h at 4 °C. Non-specific binding was determined by pre-incubating the cells with 100-fold molar excess of the non-radioactive ^nat^Ga-FAPI-46 for 5 min. After 1 h incubation at 4 °C, the cells were rinsed twice with ice-cold PBS, lysed with 1× RIPA buffer (Merck Life Science UK Limited, Dorset, UK) for 10 min and collected into scintillation vials (PerkinElmer, Beaconsfield, UK). The radioactivity was assessed using a 2480 WIZARD^2^ gamma counter (PerkinElmer, Beaconsfield, UK). The specificity of binding was calculated as percentage of added activity normalized to protein content (% ID/mg) determined by the Pierce BCA assay (Thermo Fisher Scientific) and presented as the mean of n = 3 independent measurements ± SEM. Multiple unpaired *t* tests were used for statistical analysis with GraphPad Prism version 9.0.0 (GraphPad Software, San Diego, CA, USA). Differences between groups were considered significant if *p* ≤ 0.05. Statistical significance is represented as * *p* < 0.05, ** *p* < 0.01 and *** *p* < 0.001.

For internalization studies, SF-539 cells were incubated for 30 min at 4 °C with 250 kBq/well (2 pM) of ^68^Ga-FAPI-46, and then transitioned to 37 °C for 0, 15, 30 and 60 min. After incubation, media and subsequent PBS washes were discarded, and the surface-bound radioactivity was removed by washing the cells with 400 µL of ice-cold 50 mM glycine in 150 nM NaCl (pH 3) for 5 min on ice, followed by two washes with ice-cold PBS. To obtain the internalized fraction, the cells were lysed with 400 µL of 1 M NaOH. The radioactivity in both fractions was measured by gamma counter. The percentage of internalization was calculated as
$$Internalization\left(\%\right)=\frac{internalizedfraction}{surface-boundfraction+internalizedfraction}\times 100$$

### 3.5. In Vivo Evaluation

All in vivo experiments were performed in compliance with license issued under the UK Animals (Scientific Procedures) Act 1986 and following local ethical review (project license PCC916B22, Animals in Science Regulation Unit, Home Office Science, London, UK). The studies followed the United Kingdom National Cancer Research Institute Guidelines for Animal Welfare in Cancer Research [22]. For the in vivo study, female NCR-Foxn1nu athymic mice (n = 4, 6–8-week-old, Charles Rivers Laboratories, Harlow, UK) were subcutaneously injected on the right shoulder with U87-MGvIII cells (15 × 10^4^/mouse) suspended in 30% BD Matrigel (BD Biosciences, Swindon, UK). The model was chosen based on the FAP expression and mostly on the extensive experience within the group developing GBM mice models. Tumors were allowed to grow until reaching the volume of 80 mm^3^. PET/CT imaging was performed using an Albira PET/SPECT/CT imaging system (Bruker, Billerica, MA, USA). Mice were administered ^68^Ga-FAPI-46 (65 pmol in 100 μL of 0.9% sterile saline, 2 MBq/mouse) by intravenous tail vein injection. Approximately 5 min prior to imaging, the mice were anesthetized using a mixture of isoflurane (Virbac, Bury Saint Edmunds, UK) and O_2_ (1.5–2.0% *v*/*v*) and placed in the center of the scanner’s field of view. Whole body PET static images were acquired 30–45 min post injection with a 358 to 664 keV energy window, followed by CT acquisition. The image data were normalized to correct for PET nonuniformity, dead-time count losses, positron branching ratio, and physical decay to the time of injection. No attenuation or partial-volume averaging corrections were applied. The PET images were reconstructed using a MLEM algorithm (12 iterations) with a voxel size of 0.5 × 0.5 × 0.5 mm^3^. Whole body standard high resolution CT scans were performed with the X-ray tube setup at a voltage of 45 kV, current of 400 μA and 250 projections (1 s per projection) and a voxel size of 0.5 × 0.5 × 0.5 mm^3^. The CT images were reconstructed using a FBP algorithm. Image analysis was performed using the PMOD software package (PMOD Technologies Ltd., Zurich, Switzerland). The concentration of radioactivity in the tumor was determined through volume-of-interest analysis of the PET images and the standard uptake value (SUV_mean_) was calculated by the ratio of tumor radioactive concentration and injected dose, divided by mouse weight.

For biodistribution studies, the mice were euthanized and the major organs/tissues were dissected, weighed, and the radioactivity was measured in a 2480 WIZARD^2^ gamma counter. The percentage of the injected dose per gram of tissue (% ID/g) was determined for each organ/tissue. The data are expressed as the average of n = 3 mice ± SD.

### 3.6. Ex Vivo Immunohistochemistry

Formalin-fixed tumors (10%, *v*/*v*) were embedded in paraffin, sectioned into 4 μm-thick slices, and mounted on glass slides. Multiple sequential sections were stained with hematoxylin and eosin (H&E) according to standard protocol. For all antibodies, staining was performed on an automated Leica Bond III immunostainer (Leica, Wetzlar, Germany). The following primary antibodies were used: anti-CD31 mAb (1:20, Clone SZ31, DIA310, Dianova, Hamburg, Germany), anti-Ki67 (1:100, Clone SP6, Invitrogen, Waltham, MA, USA), anti-FAP mAb (1:250, ab207178, Abcam, Cambridge, UK), and PD-L1 (Clone E1L3N, 13684, Cell Signaling Technology, London, UK). Sections were dewaxed and treated with BOND Epitope Retrieval Solution 2 (pH 9, AR9640, Leica, Wetzlar, Germany) for 10 min (CD31) or 20 min (FAP, PD-L1, Ki67) at 97 °C, followed by peroxidase blocking for 5 min at RT. Incubation with the primary antibody was performed for 15 min at RT. For CD31 staining, sections were subsequently incubated with rabbit anti-rat antibody (1:500, 312-006-045, Dianova, Hamburg, Germany) for 20 min at RT. Slides were then incubated with goat anti-rat IgG polymer detection kit (Vector Laboratories, Burlingame, CA, USA) for 8 min, followed by DAB (liquid DAB + substrate chromogen system, Dako, Agilent, Stockport, UK) for 10 min. Counterstaining was performed with Gill’s hematoxylin (Sigma-Aldrich, Taufkirchen, Germany) for 8 min. IHC and H&E stained mouse formalin-fixed paraffin-embedded (FFPE) sections were scanned on a Vectra Polaris Automated Quantitative Pathology System (Akoya Biosciences Inc., Marlborough, MA, USA) at 0.5 µm/pixel. Whole slide image analysis was performed using Phenochart, version 1.0.12 (Akoya Biosciences Inc., Marlborough, MA, USA) and HALO image analysis software, version 2.1 (Indica Labs, Albuquerque, NM, USA).

## 4. Conclusions

A new fully automated method for the preparation of ^68^Ga-FAPI-46 was successfully developed on a Trasis AiO platform. The procedure yielded the product with high radiochemical purity and apparent molar activity. Additionally, the use of a combination of HLB and QMA cartridges at the end of synthesis produced a product with high activity concentration and increased chemical purity for in vivo applications.

When tested in vitro, ^68^Ga-FAPI-46 showed high binding specificity to FAP and a quick internalization. Furthermore, the agent successfully delineated FAP-expressing tumors with high contrast images due to its increased tumor accumulation and favorable pharmacokinetics. Although our procedure for ^68^Ga-FAPI-46 preparation was developed for preclinical use, our method can be easily adapted for the clinical production and become a reliable alternative to the manual (i.e., cold kit) or modular system-based methods already described in the literature.

## Figures and Tables

**Figure 1 molecules-27-00675-f001:**
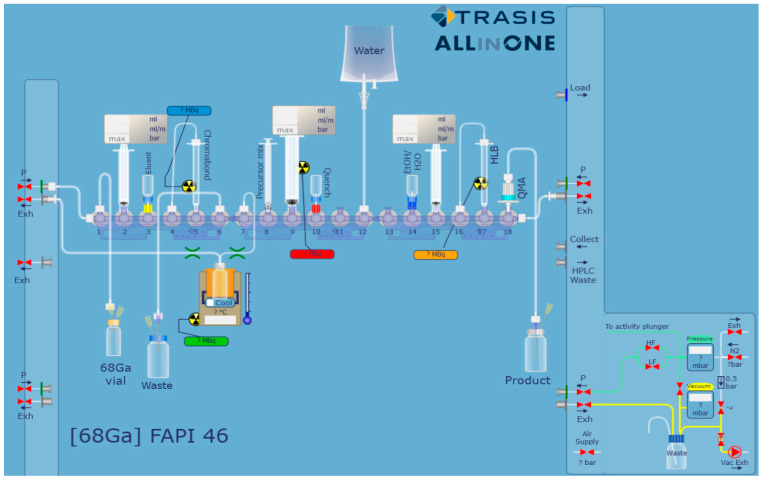
Schematic representation of the cassette set up for the preparation of ^68^Ga-FAPI-46 on a Trasis AiO.

**Figure 2 molecules-27-00675-f002:**
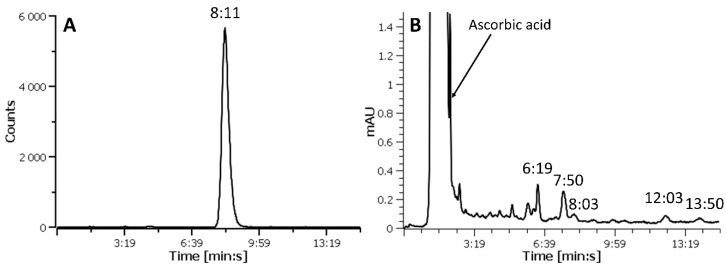
Representative HPLC radiochromatogram (**A**) and UV-chromatogram (**B**) of purified ^68^Ga-FAPI-46 showing the disappearance of the precursor (R_t_ = 8:03 min:s, Appendix A) and the presence of a variety of compounds with different elution time from the product (R_t_ = 7:45 min:s, Appendix A). The large peak eluting with the solvent front is ascorbic acid. The retention times (R_t_) are indicated as min:s.

**Figure 3 molecules-27-00675-f003:**
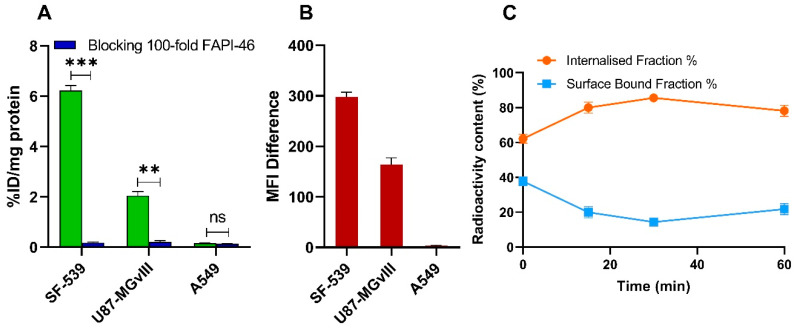
Specificity of ^68^Ga-FAPI-46 binding to SF-539, U87-MGvIII, and A549 cells (1 h, 4 °C), with or without pre-incubation with 100-fold molar excess of ^nat^Ga-FAPI-46 (**A**). FAP expression in SF-539, U87-MGvIII, and A549 cell lines determined by flow cytometry (**B**). Internalization of ^68^Ga-FAPI-46 in SF-539 cells (**C**). Data are expressed as mean ± SEM (n = 3 independent experiments). Statistical significance is represented as ** *p* < 0.01 and *** *p* < 0.001, ns = not significant.

**Figure 4 molecules-27-00675-f004:**
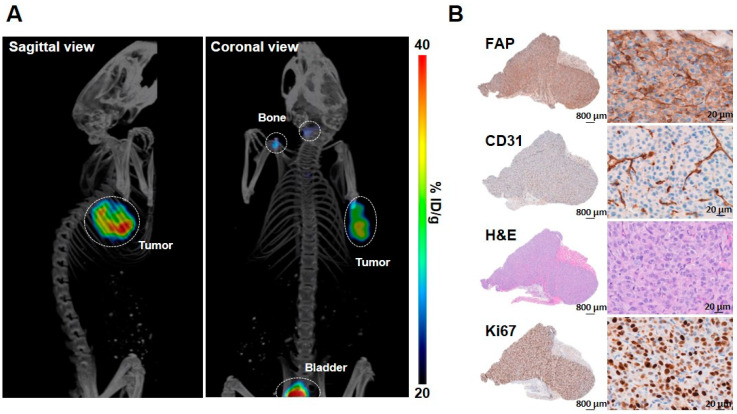
Representative PET/CT image of ^68^Ga-FAPI-46 in a mouse bearing subcutaneous U87-MGvIII tumor. Tumor, bladder and bone are outlined by a white circle (**A**). Representative U87-MGvIII tumor sections stained with FAP, CD31, HandE, and Ki67 antibodies (**B**).

**Table 1 molecules-27-00675-t001:** Summary of the process steps. The whole process lasted approximately 35 min.

1	Drawing Up of the ^68^GaCl_3_ Solution from the Delivery Vial
2	Trapping of ^68^Ga^3+^ on a SCX cartridge
3	Elution of ^68^Ga with the NaCl/HCl mixture into reactor
4	Addition of the precursor (in buffer) to the reactor
5	Incubation at 95 °C, 10 min
6	Cooling of the reactor to 40 °C
7	Removal of the reaction mixture from reactor and transfer to the HLB cartridge
8	Washing of the HLB cartridge with water
9	Elution of the HLB cartridge with ethanol-water (1:1 *v*/*v*) and passing the product solution through the QMA cartridge
10	Collection of the ^68^Ga-FAPI-46 into the product vial

## Data Availability

The data presented in this study are available on request from the corresponding authors.

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
