# Peer review of "New Fully Automated Preparation of High Apparent Molar Activity ^68^Ga-FAPI-46 on a Trasis AiO Platform"

_molecules, 2022, doi:10.3390/molecules27030675_

Round 1

Reviewer 1 Report

The paper describes an automated synthesis of the PET radiopharmaceutical 68Ga-FAPI-46 on a TRASIS module. It is well written and well crafted, with minor concerns listed below:

(1) The novelty of this work is not entirely clear. I think some effort should be spent in the introduction (or as a discussion, which I feel is missing, see also below) on explaining how this work differs from other reports on automated synthesis (two references are given in the introduction). Is the novelty only in the fact that this work uses a different module, or are some chemical improvements made? The authors should make it more clear to the reader why their work is interesting, also from a chemical perspective, especially since Molecules is not a specific radiopharmacy journal.

(2) It appears that combination of the HLB and QMA cartridges constitute a chemical improvement that was previously unknown? If this is the case, it should be made more clear. Also, it should be described in the text what types of cartridges HLB and QMA are and why they work well here, currently the uninitiated reader would have to look that up or guess. Also, a hypothesis should be given for why the specific employment of HLB-QMA in that order, as described in the text, results in higher AMA. 

(3) It is stated that the developed method can be adapted to clinical use, but it is not discussed. Please provide a discussion on what it would take for the method to see deployment in clinical production, and what potential advantages and drawbacks would be. This should include a comparison to other methods for automated production of 68Ga-FAPI on available commercial modules.

Author Response

Reviewer#1:

The paper describes an automated synthesis of the PET radiopharmaceutical 68Ga-FAPI-46 on a TRASIS module. It is well written and well crafted, with minor concerns listed below:

(1) The novelty of this work is not entirely clear. I think some effort should be spent in the introduction (or as a discussion, which I feel is missing, see also below) on explaining how this work differs from other reports on automated synthesis (two references are given in the introduction). Is the novelty only in the fact that this work uses a different module, or are some chemical improvements made? The authors should make it more clear to the reader why their work is interesting, also from a chemical perspective, especially since Molecules is not a specific radiopharmacy journal.

Response: The authors thank the reviewer for this comment.

To further clarify the importance of our work we added the following paragraph (Page 2, Introduction): “Our study provides a detailed fully automated procedure for the production of 68Ga-FAPI-46 on the extensively used commercial cassette-based synthesis platform Trasis AllinOne (AiO). The development of robust and reproducible synthetic methods on a wide variety of automated systems is a valuable tool to extend the availability and production of FAPI based radiopharmaceuticals for both preclinical and clinical application. Additionally, the new synthetic method yielded a product with reproducibly good radiochemical purity (>98%), stability and high apparent molar activity (213 ± 71 MBq/nmol) owing to the use of a combination of a polymeric-based and a strong anion exchange SPE cartridge for the purification of the final product”.

To address the final part of the comment: generally Molecules is not a radiopharmacy oriented journal but in this case the manuscript is completely on subject since it was submitted for the “Radiopharmaceuticals for PET Imaging 2021” special issue.

 (2) It appears that combination of the HLB and QMA cartridges constitute a chemical improvement that was previously unknown? If this is the case, it should be made more clear. Also, it should be described in the text what types of cartridges HLB and QMA are and why they work well here, currently the uninitiated reader would have to look that up or guess. Also, a hypothesis should be given for why the specific employment of HLB-QMA in that order, as described in the text, results in higher AMA.

Response: The reviewer is right. The purification process using HLB and QMA cartridges in sequence is a novel improvement of the previously reported automated preparations of 68Ga-FAPI derivatives. To reinforced what is stated in the Conclusions regarding the subject, a sentence to underline the novelty of the purification method has been added (Page 2, Introduction): “The new synthetic method yielded a product with reproducibly good radiochemical purity (>98%), stability and high apparent molar activity (213 ± 71 MBq/nmol) owing to the use of a combination of a polymeric-based and a strong anion exchange SPE cartridge for the purification of the final product”.

The following sentences are also added to the text (Section 2.2. Purification by SPE, page 3-4) to explain the purification system:

The HLB (Hydrophilic Lipophilic Balanced) cartridge contains a polymeric solid phase extraction material that is able to retain a wide range of compounds with different polarities. Additionally, its use allowed the recovery of the product in a small volume (approximately 400-500 µL)

The QMA cartridge contains a silica based strong anion exchange sorbent and can be used in both aqueous and non-aqueous solutions. As QMA is an anion exchanger, it will retain negatively charged compounds, such as the FAPI-46 precursor which has free carboxylic acid groups”.

This is suggesting that the order in which the reaction mixture passes through the cartridges pair is crucial to achieve an improved purification. When the reaction solution is loaded onto the HLB cartridge, 68Ga-FAPI-46 is retained while polar molecules (including ascorbic acid, acetate from the buffer, as well as free 68Ga3+) are washed off to waste. The use of an ethanol-water solution displaces 68Ga-FAPI-46 from the HLB cartridge with impurities of similar polarity. Once this solution is loaded onto the QMA cartridge, acidic and negatively charged molecules in general (such as the FAPI-46 precursor) are retained by the stationary phase while neutral components such as 68Ga-FAPI-46 passes thought the cartridge unretained and are collected into the product vial. Possibly, for the QMA cartridge to work effectively the amount of anionic compounds (such as ascorbic acid and buffers constituents) in the solution should be as small as possible and they should be removed from the product stream prior to the cartridge use. The presence of large quantities of negatively charged molecules in solution is probably the reason behind the low purification efficiency when the QMA is used before the HLB cartridge”.

(3) It is stated that the developed method can be adapted to clinical use, but it is not discussed. Please provide a discussion on what it would take for the method to see deployment in clinical production, and what potential advantages and drawbacks would be. This should include a comparison to other methods for automated production of 68Ga-FAPI on available commercial modules.

Response: The following sentence with our suggestions regarding the adaptation of our method for the clinical production of 68Ga-FAPI-46 has been added (Page 5, paragraph 2.2. Purification by SPE) as requested by the reviewer:

Because of the purification method, from the present synthetic process we expect a product with superior chemical purity compared to other reported methods. Although it was developed to produce 68Ga-FAPI-46 suitable for the preclinical use (i.e. with high AMA and activity concentration), our method can be adapted for the clinical production. Importantly, because of the availability of GMP produced pre-packaged disposable cassette and reagents, a GMP compliant production of 68Ga-FAPI-46 on a Trasis AiO platform can be achieved. In that case, the optimization of factors such as the quantity of precursor, the elution of the product from the cartridges (e.g. volume and composition of the eluent), the final product formulation and consequent stability for the determination of its shelf-life should be carried out as well as the chemical profile characterization of the final product”.

Reviewer 2 Report

The manuscript entitled “ New fully automated preparation of high molar activity 68Ga-FAPI-46 on a Trasis AiO platform” submitted by Chiara DP et. al describes an automated radiosynthesis of 68Ga-FAPI-46 on a Trasis AiO platform and evaluated its in vitro and in vivo on glioblastoma model. Please see comments below:

1) The title“ New fully automated preparation of high molar activity 68Ga-FAPI-46 on a Trasis AiO platform” mentions high molar activity but the authors have calculated AMA based on the contribution of all UV active substances in Fig. S2C. What is the molar activity using a calibration curve?

2) In Fig. 2B, which peak corresponds to natGa-FAPI-46? It seems chemically not pure. Please include a co-injection of natGa-FAPI-46 (cold standard) with purified 68Ga-FAPI-46 to confirm its identity and include in the figure 2.

3) What about the metabolic stability of 68Ga-FAPI-46 in vivo? Please perform radio metabolite analysis and discuss its results.

4) In Fig. S4 and S5, please compare with same volume (1 μl) of 68Ga-FAPI-46 at 0 min and 60 min with or without the presence of ascorbic acid? The current Fig S4 and S5 data doesn’t differentiate whether ascorbic acid is necessary. What makes the authors different when compared to reference 13 (EJNMMI radiopharm. chem. 2020, 5, 31-41) of the manuscript as the radiochemistry is known? Please give a rationale.

5) In Fig 4A, please include the scale of 0-40% ID/g to clearly visualize the difference in uptake of bone and tumor. As per Table S1, bone uptake seems to be half of tumor at 45 min post injection. Please provide a rationale for bone uptake with any evidence since it wasn’t fully understood.

6) From Fig. 2B and Fig. S3B, it is not clear the retention time of ascorbic acid unless labeled the figure S3B differently. Is 5.5 min peak in figure S3B came form the precursor? Please confirm.

7) S2 was recorded at 260 nm wavelength based on its maxima, please try at multiple wavelengths such as 220, 254 and 280 nm to see their clear UV absorbance and include in SI.

Please provide major revisions for this work.

Author Response

The manuscript entitled “ New fully automated preparation of high molar activity 68Ga-FAPI-46 on a Trasis AiO platform” submitted by Chiara DP et. al describes an automated radiosynthesis of 68Ga-FAPI-46 on a Trasis AiO platform and evaluated its in vitro and in vivo on glioblastoma model. Please see comments below:

1) The title“ New fully automated preparation of high molar activity 68Ga-FAPI-46 on a Trasis AiO platform” mentions high molar activity but the authors have calculated AMA based on the contribution of all UV active substances in Fig. S2C. What is the molar activity using a calibration curve?

Response: The authors thank the reviewer for pointing this out. We performed the molar activity determination using a calibration curve and we got values in the region of 2700-6390 MBq/nmol. However, the values were not considered very reliable since the resolution of the HPLC method was not ideal (around the Rt of the product) and the area of the UV peak of the product has been unpredictable and mostly under (or very close) to the limit of detection of our method even when large volumes of the product solution were injected. Additionally, the UV absorbing material in the final product solution is assumed to derive from the precursor molecule and therefore expected to bind competitively to FAP. The inclusion of the total UV absorbing material can better reflect the situation in the in vitro and in vivo experiments. For the reasons listed above, adding also the necessity to have quick access to MA values for the in vitro and in vivo work, we chose to consider the more accessible and reliable apparent molar activity using the total UV absorbing material instead of the vaguely estimated molar activity.

To address the reviewer comment and to be clearer and more consistent, we modified the title of the manuscript: “New fully automated preparation of high apparent molar activity 68Ga-FAPI-46 on a Trasis AiO platform”.

We also added the following paragraph in the manuscript (page 4, section “2.2 Purification by SPE”): “A molar activity (i.e. the activity per ligand, MBq/nmol) in the region of 2700-6390 MBq/nmol was calculated (using a calibration curve and considering just the area of the UV peak of the product). However, a variety of UV absorbing material was present in the final product solution (Fig. 2B and S3C). Assumed to derive from the precursor, these impurities could bind to FAP and were included in the molar activity calculation“.

Additionally, the following section was added in the caption of Fig.S3: “Molar activity (MA) values in the region of 2700-6390 MBq/nmol were calculated (by calibration curve) when only the area of the UV peak of the product (with a Rt of approximately 7:45 min:s) was taken in consideration. However, limitations of the HPLC analytical method affected the appropriate extrapolation of MA values by introducing considerable errors (i.e. unreliable integration of the UV peak of the product due to a combination of poor resolution and product peak areas that were unpredictably under or very close to the limit of detection). Additionally, a variety of UV absorbing material was present in the final product solution. Assumed to derive from the precursor and therefore expected to bind competitively to FAP, this UV absorbing material was included in the MA calculations. The apparent molar activity (AMA) of the product was thus determined and considered as a more accurate and reliable measurement than the MA.”.

2) In Fig. 2B, which peak corresponds to natGa-FAPI-46? It seems chemically not pure. Please include a co-injection of natGa-FAPI-46 (cold standard) with purified 68Ga-FAPI-46 to confirm its identity and include in the figure 2.

Response: Yes, the UV trace of the final product is not chemically pure (quite clear when zoomed in) and as the reviewer points out, a reference to the retention time of the product is missing. However, we showed the HPLC traces of both the precursor and the cold reference standard in page 3 of the Supporting Information (Fig. S2) to be used as reference as an alternative to the coinjection. To address the comment, we added the following sentence to the caption of Fig.2B: “….showing the disappearance of the precursor (Rt = 8:03 min:s, Fig. S2A) and the presence of a variety of compounds with different elution time from the product (Rt = 7:45 min:s, Fig. S2B)”.

3) What about the metabolic stability of 68Ga-FAPI-46 in vivo? Please perform radio metabolite analysis and discuss its results.

Response: As the reviewer, also the authors are of the opinion that a metabolite analysis would be necessary to have a complete picture of the radioactive agent stability in vivo. Metabolite analysis of a radiolabelled agent with a fast clearance and a very short physical half-life like 68Ga-FAPI-46 is very challenging and requires a very sensitive piece of equipment (i.e. HPLC radiodetector) that we do not possess at the moment. Additionally, as indicated by the title, the manuscript focuses on the preparation of the radioactive agent and the in vitro and in vivo studies were carried out only to test the quality of the agent we produced. We agree that a full characterisation of the radioactive agent would require metabolites analysis but we believe it is above the scope of the current manuscript and discussing the metabolic stability of the agent would deviate from the main subject of the work.

4) In Fig. S4 and S5, please compare with same volume (1 μl) of 68Ga-FAPI-46 at 0 min and 60 min with or without the presence of ascorbic acid? The current Fig S4 and S5 data doesn’t differentiate whether ascorbic acid is necessary. What makes the authors different when compared to reference 13 (EJNMMI radiopharm. chem. 2020, 5, 31-41) of the manuscript as the radiochemistry is known? Please give a rationale.

Response: We apologize for the mistake. Both experiments were carried out for 40 min (not 60 min) and this has been now corrected on the figures and caption (Fig.S5 and S6).

As the reviewer pointed out, the comparison between radiolabelling reactions carried out with or without ascorbic acid has not been investigated. In fact, to be on the safe side, the authors decided to add ascorbic acid to all reaction mixtures. The comparison was performed only on the final product (with or without addition of ascorbic acid).

The difference of injection volumes was based on the injected radioactivity amount: injecting 10µL of the solution at time 0 min would have saturated the radiodetector. While injecting 1µL of solution after 40min would have had too low counts (due to the 68 min half-life of 68Ga) to have a suitable radioactive trace. In both cases, the ROIs determination would have not been reliable. The analysis at 0 min indicated that both products had a very high (and comparable) RCP just after their preparation. The analysis after 40 min indicated that their RCP changed in a different way depending on the presence or not of ascorbic acid in the final solution.

To be clearer, the authors modified the sentence in page 5 (paragraph “2.2. Purification by SPE”) as follows: “As usually recommended, ascorbic acid was used as a radioprotectant during the radiolabeling reaction. It was added also to the purified product solution, as some degree of radiolysis was observed”.

The following reference is also added: “Shuang Liu, Charlie E. Ellars, and D. Scott Edwards, Ascorbic Acid:  Useful as a Buffer Agent and Radiolytic Stabilizer for Metalloradiopharmaceuticals, Bioconjugate Chem. 2003, 14, 5, 1052–1056”.

To address the second part of the comment: differently from the study described in reference 13, we developed and described in detail a synthetic process on a different platform (Trasis AiO instead of the Modular Lab PharmTracer and ML eazy modules), involving different set up and precursor quantity (although the description of the whole automated synthetic processes in reference 13 is not very detailed). Additionally, we improved the purification method (i.e. use of HLB-QMA cartridge combination instead of one cation exchange cartridge).

To make our rationale and novelty clearer, we added the following sentences to the manuscript, Page 2, Introduction: “Our study provides a detailed fully automated procedure for the production of 68Ga-FAPI-46 on the extensively used commercial cassette-based synthesis platform Trasis AllinOne (AiO). The development of robust and reproducible synthetic methods on a wide variety of automated systems is a valuable tool to extend the availability and production of FAPI based radiopharmaceuticals for both preclinical and clinical application. Additionally, the new synthetic method yielded a product with reproducibly good radiochemical purity (>98%), stability and high apparent molar activity (213 ± 71 MBq/nmol) owing to the use of a combination of a polymeric-based and a strong anion exchange SPE cartridge for the purification of the final product”.

5) In Fig 4A, please include the scale of 0-40% ID/g to clearly visualize the difference in uptake of bone and tumor. As per Table S1, bone uptake seems to be half of tumor at 45 min post injection. Please provide a rationale for bone uptake with any evidence since it wasn’t fully understood.

Response: We thank the reviewer for the comment since a clear understanding of the mechanisms behind the bone uptake would significantly contribute to the determination of the pharmacokinetic profile of FAPI-based agents that could further help clinical imaging planning and dosimetry. Currently, there is not much information available on this subject, even in case of 68Ga-FAPI-46 that is already used in the clinic. However, it is known that the enhanced bone uptake may be due to the presence of FAP in both murine and human multipotent bone-marrow stem/stromal cells and mesenchymal stromal cells that reside in the bone. To address the subject, the following sentence was added to the text (Page 6, section 2.4 In vivo studies):

A high bone uptake was detected (3.83 ± 1.19 %ID/g) (Fig.S8 and Table S1) similarly to previously reported studies using 68Ga labeled FAPI derivatives. This is possibly due to presence of FAP in both murine and human multipotent bone-marrow stem/stromal cells and mesenchymal stromal cells that reside in the bone”.

The following extra reference was also added: “Bae S. et al, Fibroblast activation protein α identifies mesenchymal stromal cells from human bone marrow, B. J. Heam. 2008, 142, 827-830”

Additionally, we have included a PET image with a scale 0-40% as requested. To avoid confusion, the most suitable location for the figure was considered to be in the Supplementary Information (Fig. S8 page 9).

6) From Fig. 2B and Fig. S3B, it is not clear the retention time of ascorbic acid unless labeled the figure S3B differently. Is 5.5 min peak in figure S3B came form the precursor? Please confirm.

Response: Ascorbic acid elutes with the mobile phase front and is the complex of peaks indicated by the arrow in Fig. 2B. An arrow was missing from Fig.S3B and has been added. Additionally, the following sentence has been added to Fig 2B caption for clarity: “The large peak eluting with the solvent front is ascorbic acid”.

As stated in the figure S4B (or S3B before corrections) caption, the substance eluting at 5.58 min:s is of unknown nature and might derive from the precursor. We are not sure and we did not investigate it further, therefore we did not feel confident enough to hazard a guess. It could be the same compound (or group of compounds) seen in the top HPLC trace in page 4.

7) S2 was recorded at 260 nm wavelength based on its maxima, please try at multiple wavelengths such as 220, 254 and 280 nm to see their clear UV absorbance and include in SI.

Response: The authors thank the reviewer for this comment. We agree that the analysis of the samples at various wavelengths would give a more complete and clearer picture of their compositions. We performed the analysis at 260 nm because we were mainly interested in FAPI-46 for analytical and quantification reasons and, as stated in the figure caption (page 5), what we show is figure S3 (or S2 before the corrections) is just a preliminary small study that needs further development. That would include HPLC analysis at various wavelengths, especially when the automated process will be adapted to the clinical use. The following sentence has been added to Fig S3 caption: “However, additional studies (including HPLC analysis at different wavelengths) should be performed to fully identify the impurities….” Additionally, the following sentence is added to page 5 of the manuscript (paragraph 2.2. Purification by SPE): “In that case, the optimization of factors such as the quantity of precursor, the elution of the product from the cartridges (e.g. volume and composition of the eluent), the final product formulation and consequent stability for the determination of its shelf-life should be carried out as well as the chemical profile characterization of the final product”,

Please provide major revisions for this work.

Reviewer 3 Report

The work is dedicated to the automatization of 68Ga-FAPI-46 radiopharmaceutical on Trasis AiO platform, the process was well described and the methodology is appropriate to radiopharmaceuticals synthesis method development.  The procedure could help the center that will be involved on the radiopharmaceutical application and of couse, the one that are looking on Gallium-68 automatization process.

Remarks and proposals:

HPLC's reported on supplementary material aren’t clear, could be better to split the HPLC's on multiple descriptions , to clarify and make it easy to understand, Rf are reported on description but not shows on chromatogram.
To many black lines on HPLC figures, I have try to downloaded it in different way but I still have to many black marked that make the it not easy to appreciate.

Author Response

The work is dedicated to the automatization of 68Ga-FAPI-46 radiopharmaceutical on Trasis AiO platform, the process was well described and the methodology is appropriate to radiopharmaceuticals synthesis method development.  The procedure could help the center that will be involved on the radiopharmaceutical application and of couse, the one that are looking on Gallium-68 automatization process.

Remarks and proposals:

HPLC's reported on supplementary material aren’t clear, could be better to split the HPLC's on multiple descriptions , to clarify and make it easy to understand, Rf are reported on description but not shows on chromatogram.

To many black lines on HPLC figures, I have try to downloaded it in different way but I still have to many black marked that make the it not easy to appreciate.

Response: The authors compared the original document, which clearly showed the Rt on the chromatograms, and the pdf file in the submission. It seems that the conversion from word document to pdf during the uploading of the Supplementary Information did not work properly resulting with black boxes covering some of the numbers. The authors thank the reviewer for pointing it out and apologize for the inconvenience. To avoid that again, a new format of the figures have been added to the document and the revised Supporting Information has been uploaded already in a pdf format.

Round 2

Reviewer 2 Report

1) The title“ New fully automated preparation of high molar activity 68Ga-FAPI-46 on a Trasis AiO platform” mentions high molar activity but the authors have calculated AMA based on the contribution of all UV active substances in Fig. S2C. What is the molar activity using a calibration curve?

Response: The authors thank the reviewer for pointing this out. We performed the molar activity determination using a calibration curve and we got values in the region of 2700-6390 MBq/nmol. However, the values were not considered very reliable since the resolution of the HPLC method was not ideal (around the Rt of the product) and the area of the UV peak of the product has been unpredictable and mostly under (or very close) to the limit of detection of our method even when large volumes of the product solution were injected. Additionally, the UV absorbing material in the final product solution is assumed to derive from the precursor molecule and therefore expected to bind competitively to FAP. The inclusion of the total UV absorbing material can better reflect the situation in the in vitro and in vivo experiments. For the reasons listed above, adding also the necessity to have quick access to MA values for the in vitro and in vivo work, we chose to consider the more accessible and reliable apparent molar activity using the total UV absorbing material instead of the vaguely estimated molar activity.

To address the reviewer comment and to be clearer and more consistent, we modified the title of the manuscript: “New fully automated preparation of high apparent molar activity 68Ga-FAPI-46 on a Trasis AiO platform”.

We also added the following paragraph in the manuscript (page 4, section “2.2 Purification by SPE”): “A molar activity (i.e. the activity per ligand, MBq/nmol) in the region of 2700-6390 MBq/nmol was calculated (using a calibration curve and considering just the area of the UV peak of the product). However, a variety of UV absorbing material was present in the final product solution (Fig. 2B and S3C). Assumed to derive from the precursor, these impurities could bind to FAP and were included in the molar activity calculation“.

Reviewer: Since Fig. S3D clearly indicating the impurity at 6.45 min as control reaction with FAPI-46 at 95 C for 20 min, the authors could determine this impurity by LC-MS or ESI mass since it is uv active which will give the byproduct characterization and clears the way for molar activity calculation. AMA is ambiguous in this work. The calculation of molar activity should have been revalidated.

Additionally, the following section was added in the caption of Fig.S3: Molar activity (MA) values in the region of 2700-6390 MBq/nmol were calculated (by calibration curve) when only the area of the UV peak of the product (with a Rt of approximately 7:45 min:s) was taken in consideration. However, limitations of the HPLC analytical method affected the appropriate extrapolation of MA values by introducing considerable errors (i.e. unreliable integration of the UV peak of the product due to a combination of poor resolution and product peak areas that were unpredictably under or very close to the limit of detection). Additionally, a variety of UV absorbing material was present in the final product solution. Assumed to derive from the precursor and therefore expected to bind competitively to FAP, this UV absorbing material was included in the MA calculations. The apparent molar activity (AMA) of the product was thus determined and considered as a more accurate and reliable measurement than the MA.”.

7) S2 was recorded at 260 nm wavelength based on its maxima, please try at multiple wavelengths such as 220, 254 and 280 nm to see their clear UV absorbance and include in SI.

Response: The authors thank the reviewer for this comment. We agree that the analysis of the samples at various wavelengths would give a more complete and clearer picture of their compositions. We performed the analysis at 260 nm because we were mainly interested in FAPI-46 for analytical and quantification reasons and, as stated in the figure caption (page 5), what we show is figure S3 (or S2 before the corrections) is just a preliminary small study that needs further development. That would include HPLC analysis at various wavelengths, especially when the automated process will be adapted to the clinical use. The following sentence has been added to Fig S3 caption: “However, additional studies (including HPLC analysis at different wavelengths) should be performed to fully identify the impurities….” Additionally, the following sentence is added to page 5 of the manuscript (paragraph 2.2. Purification by SPE): “In that case, the optimization of factors such as the quantity of precursor, the elution of the product from the cartridges (e.g. volume and composition of the eluent), the final product formulation and consequent stability for the determination of its shelf-life should be carried out as well as the chemical profile characterization of the final product”,

Reviewer: The authors could have done a better analysis using HPLC but it is okay to include to some extent.